# Introduction of electronic death notification in Norway—Impact on diabetes mortality registration

**Hanna M. Eng**[1]*, **Kari Anne Sveen**[2], **Stephanie Jebsen Fagerås**[2¤], **Marianne Sørlie Strøm**[2], **Lien My Diep**[3], **Petur Benedikt Juliusson**[2], **G. Cecilie Alfsen**[1,4]

**1** Department of Pathology, Akershus University Hospital, Lørenskog, Norway, **2** Department of Health Registry Research and Development, Norwegian Institute of Public Health, Oslo, Norway, **3** Oslo Center for Biostatistics and Epidemiology, Research Support Services, Oslo University Hospital, Oslo, Norway, **4** Faculty of Medicine, University of Oslo, Oslo, Norway

☯ These authors contributed equally to this work.
¤ Current address: Capgemini, Insights & Data, Bergen, Norway
* hanna_maria_e@hotmail.com

**Data Availability Statement:** All relevant data are within the manuscript and its Supporting Information files.

## Abstract

### Introduction

We studied changes in death statistics by deaths from diabetes mellitus (DM) after introduction of mandatory online death certificate (DC) submission in Norway.

### Materials and methods

Information on deaths with DM mentioned in the DCs from year 2017 (DCs submitted on paper) to 2022 (DCs submitted online) was collected from the Norwegian Cause of Death Registry (NCoDR), Sex, age, year of death and type of DC (paper (pDC) vs electronic (eDC)) was registered. In DCs with DM as underlying cause of death (UCOD), all codes (International classification of diseases, 10th revision (ICD-10)), their original position in the DC and place of death were collected. DM was classified as type-1, type-2 and other. Differences between 2017 and 2022 according to use of unspecified DM diagnoses, number of changed diagnoses after automated processing, correct positioning of UCOD in DC, total number of diagnoses, and use of ill- defined diagnoses were analyzed. Generalized linear models for binomial outcome with log link were used to fit mortality data and test differences between electronic and paper registration systems, two-sample t-test and linear regressions for analysis of differences in number of diagnoses.

### Results

229 807 deaths were registered, including 3 864 DM deaths. Online DC submission increased from 0 in 2017 to 95% in 2022. In 2022, DCs with DM as UCOD showed significant less use of unspecified diabetes diagnoses (Relative risk,RR: 0.18, 95% confidence interval (CI): 0.14–0.22), reduced need for change of diagnoses after automated processing (RR: 0.52, CI: 0.46–0.59), reduced number of diagnoses (CI: -0.7 to -0.38), and less use of ill-defined diagnoses (RR: 0.83, CI: 0.71–0.97).

**Funding:** The author(s) received no specific funding for this work.

**Competing interests:** The authors have declared that no competing interests exist.

## Conclusions

The introduction of online cause of death submission in Norway improved the quality of registration of deaths from diabetes.

## Introduction

Cause of death certificates (DC) are cornerstones in the cause of death statistics, which in turn is of crucial importance for monitoring public health. DCs in Norway are issued by physicians, who until 2018 completed the certificates manually, with hand-written diagnoses in free text on paper forms (pDC) [1]. After signing, the physicians forwarded the paperforms to the Norwegian Cause of Death Registry (NCoDR), by means of a postal route with at least two intermediate steps, involving the district courts for registration and the local municipal doctor [2]. The paper-based system was time-consuming and required a ten month elapse before the final statistics for a cohort could be published, and up to 22 months from time of death until cause of death appeared in the statistics [2]. Not least from the perspective of health monitoring and preparedness, the possibility of faster processing through use of electronic solutions is of major advantage. Other benefits of online solutions, as clear writings and unmistakable signatures, the possibility of immediate feedback, use of stepwise guidance or built-in restrictions and drop-down menus may also involve major improvements to the death certification system.

Thus, the possibility of online submission has been introduced in several countries [3–7]. However, few countries have electronic submission as mandatory [3,6]. Denmark was the first country in the world where a fully electronic reporting system of death was introduced, in 2007 [3].

From September 2018, an online electronic version (eDC) became available in Norway [8]. From 1st of January 2022, the use of eDC became mandatory. No targeted training of the physicians was carried out in connection with the introduction of the new submission system. A preexisting, voluntary online course on death certification offered by the Norwegian Medical Association, was updated and instructions about the new online registration system was published on the web-pages of The Norwegian Directorate of Health and Norwegian Institute of Public Health [9–11].

Information on causes of death in both pDCs and eDCs are structured in two parts [12]. Part I contains the "chain of events", starting with the immediate cause (Ia) and going stepwise back to the underlying cause of death (UCOD), defined as the illness or injury that started the morbid condition leading directly to death [12]. For each death, only one UCOD should be given. Part II contains contributing causes, defined as other significant conditions that may have contributed to death, but without being directly or causally related to the condition that caused death.

The structure of the Norwegian eDC was introduced with few modifications: The chain of events in part I was expanded one step, from a-c to a-d, according to recommendations from WHO [12]. The possibility of free text at all steps in part I was kept unchanged, but a drop-down menu with diagnoses from ICD-10 appears automatically while writing, simplifying the physicians selection of causes of death. If using the drop-down menu in part I, only one choice is allowed per step.

After submission, the processing of DCs by the NCoDR remains unchanged. eDCs are automatically assigned with ICD-10 codes by means of the automated coding system (Iris), based on the Automatic Classification of Medical Entry software (ACME) [13,14]. If the death

certificate contains free text, and the automated processing is not able to allocate an ICD-code, diagnoses are registered manually. Subsequent handling by Iris results in a "string of codes", reflecting the order of diseases or conditions given in the DC. The string of codes is not affected by type of submission and/or use of free text or drop-down menu. The string of codes from each DC is processed by Iris/ACME, which selects UCOD according to rules and guidelines provided by WHO (ICD-10) [15]. Automated processing may lead to changes or specifications of the original ICD-10 code on the DC, dependent on the setting of diagnoses on the DC. For instance will deaths from type 1 diabetes without complications (E10.9) imply a change to type 1 diabetes with renal complications (E10.2) if chronic renal failure or diabetic nephropathy is mentioned elsewhere in the code string.

Diabetes mellitus causes yearly less than 2% of deaths in Norway [16]. Although relatively few in numbers, a distinct increase in diabetes deaths was registered from 2020 [17]. The increase, which was also seen internationally, coincided both with the SARS-CoV-2 pandemic and the introduction of the online cause of death registration system [18–20]. Changes in type of registration system may influence the cause of death statistics, but studies on the impact of changes due to the introduction of online registrations are limited. Being a disease with distinct changes in mortality in the current time period, and with a wide range of diagnostic possibilities, diabetes mellitus was chosen to assess the impact on certification quality and cause of death statistics by introduction of an electronic cause of death registration system in Norway.

## Materials and methods

### Data

The study is an observational, population-based registry study. We extracted data from the NCoDR on all deaths in Norwegian citizens 01.01.2017–31.12.2022, included Norwegian citizens who died abroad, but not foreign citizens who died in Norway. As autopsy may influence the registration of cause of death, and we wanted to study the effect of eDC, all deaths with autopsy were excluded.

We extracted information on all causes of deaths and deaths with diabetes-related conditions (WHO ICD-10: E10.0-E14.9) on DCs [21].

In deaths with DM as UCOD, information on all ICD-10 codes (code string), their original position on the DC (part I a-d or part II), type of DC (paper vs electronic), place of death (hospital, other health institutions including nursing homes, private home, unknown), age, sex and year of death were extracted. In deaths with DM present in the code string but not specified as UCOD, the position of the DM-codes was registered. Diabetes was classified as DM type-1 (T1DM, ICD-10: E10.0-E10.9), DM type-2 (T2DM, ICD-10: E11.0-E11.9) and DM-other (ICD-10: E12.0–12.9: malnutrition-related DM, E13.0–13.9: other specified DM, E14.0–14.9: unspecified DM).

### Quality assessment

We decided for five quality measures for changes in certification quality before (2017) and after (2022) introduction of eDCs: 1) number of UCODs grouped as DM-other, and 2) number of DM UCODs subjected to change after automated processing, as variables for diagnostic precision, 3) number of DM UCODs with correct positioning in part I, as indicator for structural improvement/understanding, 4) number of diagnoses, as indicator for information quantity, and 5) number of ill-defined diagnoses, as indicator for information quality.

Correct position in part I (quality variable 3) was only counted from DCs with no changes in UCOD after automated processing. Number of diagnoses (variables 4 and 5) were counted from the code strings, i.e. from both parts I and II. Ill-defined conditions (variable 5) were

defined as all codes in chapter XVIII (symptoms, signs and abnormal clinical and laboratory findings, not elsewhere classified, R00-R99), I46.9 (cardiac arrest, unspecified), I95.9 (hypotension, unspecified), I99 (other and unspecified disorders of circulatory system), J96.0 (acute respiratory failure), and J96.9 (respiratory failure, unspecified).

### Statistical methods

Age-standardized mortality rates (ASMR) on deaths without autopsy were computed by the direct standardization method, using 5-year age strata and the European Standard Population of 2013 [22]. Generalized linear models (glms) for binomial outcome with log link were used to fit mortality data and test whether the proportion of diabetes deaths among total number of deaths of any cause differed between electronic and paper registration system adjusting for year of death, age and sex. The analyses were performed for diabetes in general and stratified by diabetes types. Differences between the adjusted proportions of the electronic and paper registration of death were reported as relative risks (RR) with 95% confidence intervals (CI) and p-values provided for additional information. The variable for year of death was centered and the age groups with small numbers of death were combined before fitting the glms. For estimation of changes in quality variables, only number of DM deaths in the years 2017 and 2022 were included in the analyses. Due to small numbers of pDCs in 2022, the total number of DM deaths was used in the calculation. Generalized linear model with log on link for binary outcomes (i.e the same method for comparing pDCs and eDCs) were used to estimate RR and CI for the quality variables 1, 2, 3 and 5. Total number of ICD-codes (variable 4) were normally distributed. Therefore, a two-independent sample t-test and linear regressions were used to estimate additive changes and CI. The results for variable 4 and 5 were adjusted for sex and age. The analyses were performed with Stata/SE version 17.0 and 18.0 for Windows. Relative changes were calculated for some of the data.

### Ethical approval

The study was defined as a quality study within the Norwegian Cause of Death Registry and ethical approval was thereby not required [23].

### Results

A total number of 250 298 deaths were registered in the study period (2017–2022). We excluded 20 491 autopsied cases from further analyses, including 299 deaths with DM as UCOD. A total of 229 807 DCs were included in the study.

### Death rates

Only minor variations in the number of deaths were seen from 2017 until 2021. From 2021 to 2022, during the second year of the SARS-CoV-2 pandemic, number of deaths showed a 7% relative increase (CI: 1.10–1.13, p <0.001) (**Table 1**).

Age standardization confirmed the excess mortality in 2022, with ASMR increasing from 2021 to 2022 with 11.9% in men and 7.1% in women (**S1 File**). Death from DM increased already markedly in 2020, coinciding with the first year of the SARS-CoV-2 pandemic [24] (**Fig 1**).

Increased ASMR was seen every year from 2019 for both T1DM and T2DM. From 2019 to 2022, ASMR for T1DM increased from 1.2 to 2.8 in men (relative increase 133%) and from 0.8 to 1.5 in women (relative increase 88%). For T2DM, ASMR in men and women increased

**Table 1. Number of deaths in Norway in 2017–2022, according to sex, age and type of death certificate submission.**

| | | 2017 | 2018 | 2019 | 2020 | 2021 | 2022 |
|---|---|---|---|---|---|---|---|
| **All** | N | 37270 | 37386 | 37160 | 37271 | 38169 | 42551 |
| | Mean age Y (SD) | 81.2 (13.1) | 81.3 (12.9) | 81.3 (13.0) | 81.3 (12.8) | 81.6 (12.6) | 81.5 (12.7) |
| **Male** | N (%) | 17424 (47) | 17587 (47) | 17636 (47) | 17795 (48) | 18012 (47) | 20669 (49) |
| | Mean age Y (SD) | 78.6 (13.3) | 78.8 (13.0) | 78.9 (13.0) | 79.2 (12.6) | 79.4 (12.5) | 79.3 (12.7) |
| **Female** | N (%) | 19846 (53) | 19799 (53) | 19524 (53) | 19476 (52) | 20157 (53) | 21882 (51) |
| | Mean age Y (SD) | 83.4 (12.6) | 83.5 (12.5) | 83.3 (12.6) | 83.3 (12.6) | 83.5 (12.3) | 83.5 (12.3) |
| **Type of submission** | N (pDC) (%) | 37270 (100) | 37316 (99.8) | 35970 (96.8) | 23418 (62.8) | 8007 (21.0) | 1995 (4.7) |
| | N (eDC) (%) | 0 | 70 (0.2) | 1190 (3.2) | 13853 (37.2) | 30162 (79.0) | 40556 (95.3) |

Deaths with autopsies are excluded. N; number, Y; years, SD; standard deviation, pDC; paper death certificate, eDC; electronic death certificate.

from 7.6 to 16.8 (relative increase 121%) and from 4.5 to 9.0 (relative increase 100%), respectively (**S1 File**).

The group of DM-other consisted mainly of deaths from unspecified diabetes (E14.0–14.9), which constituted close to 100% of UCODs in this group from 2017–2020 (all but one case),

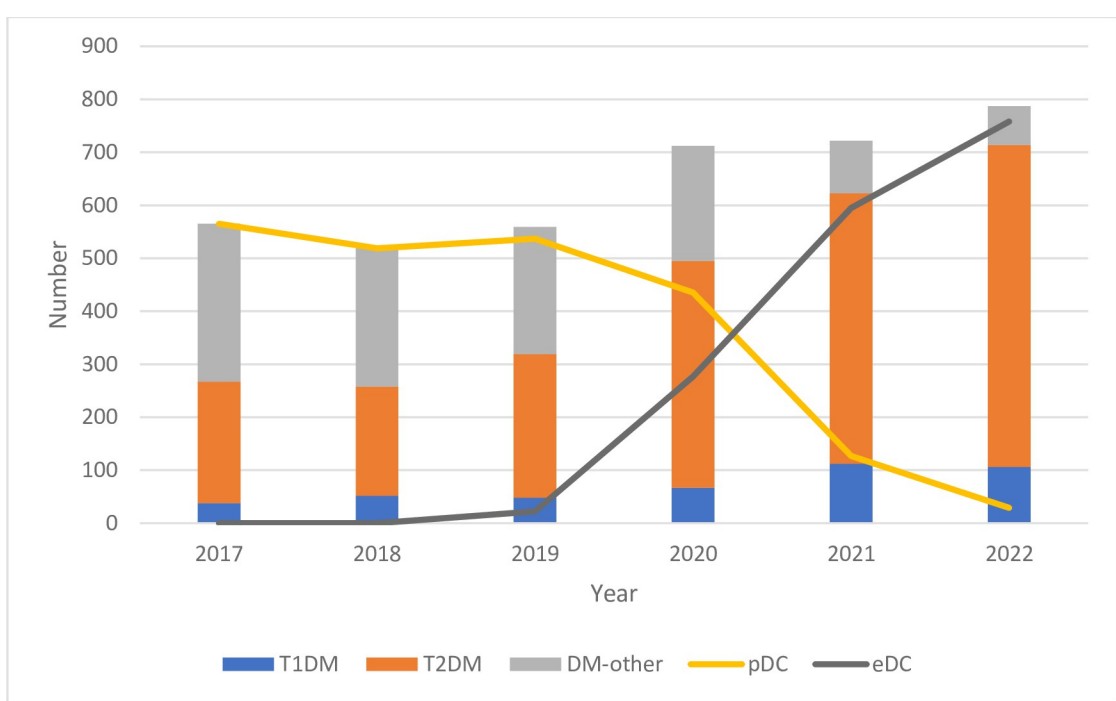

**Fig 1. Number of deaths with diabetes mellitus (DM) as underlying cause of death in 2017–2022, according to type of DM and death certificate submission.** Deaths with autopsy are excluded. T1DM; Diabetes type-1, T2DM; diabetes type-2, pDC; paper death certificate, eDC; electronic death certificate.

95% in 2021 and 82% in 2022 (**S2 and S6** Files). Especially in women, mortality rates for DM-other showed a downward trend already from 2017, but were reduced in both sexes after 2020, with > 50% reduction from 2020 to 2021, and with a reduction from 2017 to 2022 of 44% (relative reduction 83%).

## eDC coverage

In 2022, the first year of mandatory online submission, 95% of all DCs in Norway were submitted electronically (**Fig 1 and Table 1**).

Death from DM was significantly more often submitted electronically than other causes of death when adjusted for year of death, age groups and sex (RR = 1.15, CI: 1.03–1.28, p = 0.011), but with differences for the different types of diabetes: T1DM and T2DM were submitted electronically significantly more often than in death from all causes (T1DM: RR = 1.97, CI: 1.41–2.75, p<0.001, T2DM: RR = 2.04, CI: 1.77–2.36, p<0.001), while death from DM-other more often was submitted manually, on paper (RR = 0.15, CI: 0.12–0.20, p<0.001).

Transition to eDC was slower for deaths in private homes (**S3 File**). In 2022, 8% of all DM deaths in private homes were still submitted as pDC, in contrast to 2% of DM deaths in hospitals and in other institutions. Also, the diagnosis of unspecified diabetes (DM-other) was more commonly used for deaths in private homes. In 2022, DM-other constituted 15% of all DM-deaths in private homes, in contrast to 8% of DM-deaths in hospitals and other institutions.

## Quality assessments

Quality changes from before and after introduction of online DC submission are summarized in **Table 2**.

Quality variables 1, 2, 4 and 5 underwent significant changes after introduction of mandatory online DC submission in 2022 when compared to year 2017, which was the last year with all DCs submitted on paper. Diagnostic precision increased, assessed by the reduction in

**Table 2. Quality variables in death certificates (DCs) with diabetes mellitus (DM) as underlying cause of death (UCOD), before (2017) and after (2022) introduction of electronic submission.**

| Variable | 2017 | 2022 | Statistics |
|---|---|---|---|
| 1) UCOD as DM-other, N (%) | 298 (53) | 73 (9) | RR: 0.18 CI: 0.14–0.22 P<0.05 |
| 2) UCOD changed after automated processing, N (%) | 365 (65) | 265 (34) | RR: 0.52 CI: 0.46–0.59 P<0.05 |
| 3) UCOD positioned in part I, N (%) | 116 (58) | 334 (64) | RR: 1.10 CI: 0.96–1.26 P = 0.152 |
| 4) Diagnoses per DC, mean number | 4.52 | 4.0 | CI: -0.7 to -0.38 P<0.05 |
| 5) Ill-defined diagnoses per DC, mean number | 0.52 | 0.43 | RR: 0.83 CI: 0.71–0.97 P = 0.018 |

N, number, RR, relative risk; CI, confidence interval.

unspecified DM-diagnoses (DM-other) and the need for changes after automated processing of the diagnoses. The slight, but not significant increase of DCs with UCODs correctly positioned in part I could be a result of the structure of the eDC, with online guidance through the step-by-step completion. The new step Id was used for the first time in 2019 and constituted 13% of all DM UCODs located in part I in 2022 (**S4 File**). The simultaneous use of DM diagnoses in part I and II was rare. From 2017–2022, 45 DCs had DM mentioned twice, and most often in pDCs (28 cases) (**S5 File**).

Quantity and quality of information, as measured by the number of diagnoses and ill-defined diagnoses in the code strings were also reduced, with relative reductions of 11% and 17%, respectively.

## Non-diabetes deaths

In deaths from non-diabetes causes, the mentioning of DM as one of the diagnoses increased with 4% in the study period, which is little compared to the 39% increase of DM as UCOD in the same period. The majority of DM diagnoses appeared as expected in part II, but with a downward trend. Correct positioning of DM as a contributing disease in part II was reduced from 74% in 2017 to 68% in 2022, a relative reduction of 8% (**Table 3**). The reduction may be explained by an increased use of position Id in online submitted DCs. When excluding cases with DM in Id (n = 159) in 2022, no reduction of DM as contributing disease in part II is found, when compared to 2017 (73%).

## Discussion

### Key results

After introduction of electronic submission of DCs in Norway, the quality of cause of death reporting regarding deaths due to diabetes mellitus was improved. UCOD in diabetes deaths was more precise, with less use of unspecified DM diagnoses and a reduced number of changed diagnoses after automated processing. Also, the number of diagnoses in the DCs were reduced, and ill-defined diagnoses were significantly less used.

### Limitations by the pandemic

Shortly after the introduction of eDC in 2019, the world was hit by the SARS-Cov2 pandemic. Apart from deaths due to COVID-19, the pandemic also influenced non-COVID-19 mortality

**Table 3. Death certificates (DCs) with non-diabetes underlying cause of death and diabetes mellitus (DM) mentioned on DCs, according to position of DM diagnoses.**

|  | 2017 | 2018 | 2019 | 2020 | 2021 | 2022 |
|---|---|---|---|---|---|---|
| **DM mentioned, N** | **2252** | **2209** | **2246** | **2387** | **2209** | **2345** |
| **Part I, N** | 591 | 533 | 588 | 685 | 685 | 755 |
| Ia | 39 | 22 | 27 | 39 | 47 | 57 |
| Ib | 274 | 261 | 273 | 299 | 253 | 252 |
| Ic | 277 | 249 | 286 | 296 | 287 | 287 |
| Id | 1 | 1 | 2 | 51 | 98 | 159 |
| **Part II, N** | 1661 | 1676 | 1658 | 1702 | 1524 | 1590 |
| **Part I and II, N** | 31 | 16 | 12 | 6 | 21 | 29 |

Deaths with autopsy are excluded. N; number. Death certificates with DM in both parts I and II are counted only once, in position I. Position Id in 2017 is created by handwriting.

[25,26]. Diabetes mellitus (DM) was one of the disease groups that showed signs of changing mortality in many countries, including Norway [17–20]. No increase of DM has been reported as a result of change in DC submission type. Statistics from Denmark, where eDC was introduced in 2007, showed no increase in DM mortality in the years after introduction [27]. Thus, we are confident that the increase of DM from 2020 in Norway is mainly due to the pandemic and not a result of change in cause of death submission.

The transition from paper to online submission was accelerated by the pandemic. Institutions were strongly recommended by the Norwegian Institute of Public Health to report COVID-19 deaths electronically to support timely surveillance of these deaths. This resulted in not only a high coverage of electronically reported COVID-19 deaths, but of all causes of death. The acceleration was also driven forward by the decision of mandatory eDC registration from 1st of January 2022. A higher effectiveness in transition to online registration was achieved in both Denmark and Portugal, where online DC submission became mandatory in 2007 and 2014 respectively. Within 2 years 100% coverages were achieved, without the drive from a pandemic [6,28,29]. Without mandatory implementation, the transition to online systems remains slow. After 3 years of voluntary use in France, the coverage was still only 5% [30].

No targeted education in online submission was carried out as it was expected the electronic system would be self-explanatory. It remains to be seen to what extent this has been successful. It will in any case mean a quality improvement to introduce mandatory courses in death certificate submission. The pandemic led to an increased focus on cause of death in general, and probably caused an increased awareness and knowledge among certifying physicians about the construction of the "chain of event". However, this effect is expected to be transitory.

## Extension of the chain of events, Id

The electronic submission system introduced an extra step d in part I. A similar change in the chain of events in USA in 1988 was claimed to have caused an increase of DM deaths in the following year, although the connection was never proven [31]. We are not aware of similar changes elsewhere. The use of position Id in the chain of events in our study was too rare to influence the diabetes rates. However, part Id was increasingly used for DM-diagnoses in non-diabetes deaths, while correct positioning in position II decreased correspondingly. The erroneous use of Id in non-diabetic deaths probably reflects a structural problem of the eDC-version, leading to misunderstandings about positioning of contributing conditions.

## Drop-down menu

A more significant change in online submission was the introduction of a drop-down menu, enabling registration without adding free text and allowing only one drop-down menu diagnose per step. A possible negative effect by a drop-down menu may be the introduction of a bias towards the diagnoses first appearing, the so-called position bias [32]. To minimize this effect, all diagnoses and codes in the relevant ICD-chapter is shown when the drop-down menu appears. The fact that T2DM-diagnoses, which appear "below" T1DM-diagnoses in the drop-down menu, were used more often than T1DM, speaks against position bias.

We have not been able to register the extent of the use of the drop-down menu. The reduction in number of diagnoses given in eDCs may however be the result of drop-down menu use, allowing only one diagnose per step. The combination of text and codes in drop-down menus with suggestions appearing while writing, a system similar to the drop-down menu in Norway, has been found to improve clinical information in electronic health records

significantly [33]. In the present study, we had no access to medical records, and were thus unable to conclude on any diagnostic improvements due to online submission.

## Number of diagnoses

Lefevre et al compared the quality of paper versus electronic death certificates after transition to eDC in parts of France, and found an increase in ICD-10 codes and a reduction of ill-defined diagnoses [5]. The French study differs from the present as it included death from all causes, not only DM. The reduction of diagnoses/ICD-10 codes in the code string from eDCs with DM deaths in the present study may be due to the allowance of only one diagnosis per step in part I when using the drop-down menu. We have no information of a drop-down menu in the French eDC system.

Similar to Lefevre et al, we did find a significant reduction in the use of ill-defined diagnoses after introduction of eDC. The level of ill-defined diagnoses is used as one of the quality measures in cause of death statistics [34]. Further studies are needed to prove if the reduction of ill-defined diagnoses on Norway also applies to other causes of death than DM.

## Conclusions

Online cause of death submission clearly improves the quality of registration of deaths from diabetes. This may indicate that online submission may improve the quality of death statistics also for other causes of death.

## Supporting information

**S1 File. ASMR per 100 000 in Norway 2017–2022.**
(PDF)

**S2 File. DCs with DM as UCOD, specified type of DM, type of death certificate submission, sex and mean age.**
(PDF)

**S3 File. Deaths by DM, according to place of death, type of diabetes and type of submission.**
(PDF)

**S4 File. DM as UCOD after automatic processing of diagnoses from DCs, according to use of original or changed diagnoses.**
(PDF)

**S5 File. DM as UCOD from DCs, according to type of death certificate and the original position of diabetes type diagnoses.**
(PDF)

**S6 File. Diabetes ICD-10 codes used in cases with diabetes mellitus (DM) as underlying cause of death.**
(PDF)

## Author Contributions

**Data curation:** Stephanie Jebsen Fagerås, Lien My Diep.

**Formal analysis:** Stephanie Jebsen Fagerås, Lien My Diep.

**Methodology:** Hanna M. Eng, Kari Anne Sveen, Stephanie Jebsen Fagerås, Marianne Sørlie Strøm, Lien My Diep, Petur Benedikt Juliusson, G. Cecilie Alfsen.

**Software:** Stephanie Jebsen Fagerås, Lien My Diep.

**Supervision:** Kari Anne Sveen, Marianne Sørlie Strøm, Petur Benedikt Juliusson, G. Cecilie Alfsen.

**Writing – original draft:** Hanna M. Eng, G. Cecilie Alfsen.

**Writing – review & editing:** Hanna M. Eng, Kari Anne Sveen, Stephanie Jebsen Fagerås, Marianne Sørlie Strøm, Lien My Diep, Petur Benedikt Juliusson, G. Cecilie Alfsen.

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
