## [Decision Letter · Decision Letter 0]

11 Mar 2024

PONE-D-24-05293Introduction of Electronic Death Notification in Norway - Impact on Diabetes Mortality RegistrationPLOS ONE

Dear Dr. Eng,

Thank you for submitting your manuscript to PLOS ONE. After careful consideration, we feel that it has merit but does not fully meet PLOS ONE’s publication criteria as it currently stands. Therefore, we invite you to submit a revised version of the manuscript that addresses the points raised during the review process.

We look forward to receiving your revised manuscript.

Kind regards,

Pasyodun Koralage Buddhika Mahesh

Academic Editor

PLOS ONE

Reviewers' comments:

Reviewer's Responses to Questions

**Comments to the Author**

1. Is the manuscript technically sound, and do the data support the conclusions?

Reviewer #1: Yes

Reviewer #2: Partly

Reviewer #3: Partly

2. Has the statistical analysis been performed appropriately and rigorously? 

Reviewer #1: Yes

Reviewer #2: No

Reviewer #3: No

3. Have the authors made all data underlying the findings in their manuscript fully available?

Reviewer #1: Yes

Reviewer #2: No

Reviewer #3: No

4. Is the manuscript presented in an intelligible fashion and written in standard English?

Reviewer #1: Yes

Reviewer #2: Yes

Reviewer #3: Yes

5. Review Comments to the Author

Reviewer #1: This study has been conducted in Norway to study the impact of introducing the e death certification system in 2018 and making the e notifications mandatory from 2022. The authors have compared the diagnosis of Diabetes as the underlying cause of death (UCOD) from 2017 to 2022. The study shows in increase of reporting diabetes as UCOD after the introduction of e notification in 2018 and becoming e DC mandatory in 2022.

The rationale for the study is clear and valid. The authors have used a technically suitable protocol and a feasible methodology to achieve the study's aims effectively. I congratulate the authors for doing this all-important study after a change in the death certification system.

However, I have the following comments about their methods and the study.

1. In a paper-based certification system, the doctors will enter cause of death data in free text in parts 1 and 2 of the WHO recommended format in handwritten format. In an e death certification system doctors will enter free text in the same WHO recommended format but the causes are type written. The main advantages are clarity due to lack of illegible handwriting, adherence to the prescribed format, and following the guidelines (e.g. One condition per line, sequence, mandatory time intervals, etc.) In this study, the authors do not discuss the immediate advantages of e-certification.

2. The main benefits the authors discuss are not due to e-certification but to introducing a drop-down menu of ICD-10 categories and codes. Drop-down menus are known to cause systematic errors in certification and coding. Using a drop-down menu, the certifiers bypass volume 3 of ICD-10 and directly assign ICD codes from the tabular list, bypassing exclusion criteria and other coding conventions in volume 1. This is an inherent error in using drop-down menus.

3. The authors have not described how the coding was done before the certification system. It may be Iris's text entry mode based on a dictionary. With the introduction of e certification, now they use Iris in the code entry mode. But the primary code assignments to individual causes of death from the drop-down menu may not be the best code choices for assignment.

4. Reduction of ill-defined ICD codes is a positive finding. However, the increase in specific diabetes mellitus codes does not necessarily indicate improved certification and coding practices because the codes are necessarily from the drop-down menu.

5. If the study's objective is to investigate the certification practices after e certification system, the authors must conduct a medical record review to validate e death certificate findings with the hospital medical record content. If such a validation proves an increased sensitivity and Positive Predictive Value, it will be a proof of improved cause of death data following e certification.

6. I would suggest the authors to screen death certificates for the reduction of common errors in medical certification. The most common errors are,

a. Reporting multiple causes per line

b. Absence of time intervals

c. Illegibility

d. Use of non-standard abbreviations

e. Reporting ill-defined causes and modes of dying as underlying cause of death

f. Incorrect causal sequence in part 1 of the death certificates

g. Etc.

7. The authors can still study the impact on diabetes mortality registration. However, while doing that you may identify the drop-down menu in e certificate as a limiting factor.

Reviewer #2: This article addresses an important topic - that of the impact of the introduction of electronic death notification on reporting of diabetes mortality. However there are several serious deficiencies in how the study is presented that need to be addressed if it is to be suitable for publication.

The main issue is that many of the results that are described in the Abstract and the main text - specifically the relative risks from the generalized linear models - are not presented in any tables. No Table is referenced in the main text and the results are not presented in any of the tables that are shown. This means that it is not possible to assess the results adequately. Presentation of such results in a table is a pre-requisite for such a manuscript.

The manuscript does not adequately provide a rationale for the study and the knowledge gap that the study is filling. Why is accurate reporting of diabetes important? For example, what is its ranking as a leading cause of death or disease burden in Norway, what is its impact on the health system? There also needs to be more detailed exploration of the literature on this topic and description of the existing knowledge gap, so that the reader understands the contribution that the manuscript makes.

Is there a variable about region of the country that could be included in the models?

There also needs to be more background about the introduction of the electronic death certificate - how it was rolled-out, how doctors were trained, etc - as well as the process of a doctor completing the electronic death certificate and submitting the certificate.

What does it mean by "variable for year of death was centered"? This should be clearer.

The information in Lines 159-163 is not really important and can be deleted or significantly reduced.

The writing needs to be improved. There are several places with terminology that needs to be more detailed (Line 31 "International classification of diseases 10 codes" should be "10th Revision") or there are errors (Line 36 eCD instead of eDC), I44.9 for cardiac arrest should be I46). Please review carefully.

What is "Id" in Line 240?

Line 232: "January 1st2022" should be corrected.

Reviewer #3: While commend the efforts of the authors, the following comments are given with the hope that these will be beneficial for them.

Major comments:

1. The introduction section needs to be enriched with more details; more information of the paper-based system used, why countries are shifting to electronic systems, brief account on what is meant by quality of mortality registration and data on diabetes burden (e.g. prevalence) in the Norway.

2. It would be better to clearly mention and operationalize the variables assessed for quantity and quality in one place in the methodology.

3. More details are needed in the methods section on how the data analyses were done. As an example how were verification done (e.g. whether two investigators analyzed the data independently etc.)

4. It would be better if a table can be inserted to show the relative risk statistics and p values mentioned in abstract and in results.

Minor comment

-There is a confusion on the uniformity of the abbreviations used. Examples include:

1. What is meant by "CDs" in the last sentence of the results in abstract

2. eCD and eDC in methods of abstract

6. PLOS authors have the option to publish the peer review history of their article (what does this mean?). If published, this will include your full peer review and any attached files.

Reviewer #1: **Yes: **U S H Gamage

Reviewer #2: No

Reviewer #3: **Yes: **I.O.K.K.Nanayakkara

---

## [Author Response · Author response to Decision Letter 0]

25 Apr 2024

Response to reviewers:

Reviewer #1: This study has been conducted in Norway to study the impact of introducing the e death certification system in 2018 and making the e notifications mandatory from 2022. The authors have compared the diagnosis of Diabetes as the underlying cause of death (UCOD) from 2017 to 2022. The study shows in increase of reporting diabetes as UCOD after the introduction of e notification in 2018 and becoming e DC mandatory in 2022.

The rationale for the study is clear and valid. The authors have used a technically suitable protocol and a feasible methodology to achieve the study's aims effectively. I congratulate the authors for doing this all-important study after a change in the death certification system.

However, I have the following comments about their methods and the study.

1. In a paper-based certification system, the doctors will enter cause of death data in free text in parts 1 and 2 of the WHO recommended format in handwritten format. In an e death certification system doctors will enter free text in the same WHO recommended format but the causes are type written. The main advantages are clarity due to lack of illegible handwriting, adherence to the prescribed format, and following the guidelines (e.g. One condition per line, sequence, mandatory time intervals, etc.) In this study, the authors do not discuss the immediate advantages of e-certification.

Information about the main and immediate advantages of an electronic death certification system are now incorporated in the Introduction, see lines 65-69 in the manuscript.

2. The main benefits the authors discuss are not due to e-certification but to introducing a drop-down menu of ICD-10 categories and codes. Drop-down menus are known to cause systematic errors in certification and coding. Using a drop-down menu, the certifiers bypass volume 3 of ICD-10 and directly assign ICD codes from the tabular list, bypassing exclusion criteria and other coding conventions in volume 1. This is an inherent error in using drop-down menus.

The drop-down menu in the Norwegian online system allows searches both by text and codes, as text (with codes) appears while writing, thus preventing bypassing of volume 3 of the ICD-10, a possible bias discussed by reviewer #1. We have clarified the advantages and the structure of the drop-down menu in the Introduction and expanded the discussion on any negative effects in the Discussion. See lines 66-69 and 285-299 iin the manuscript.

3. The authors have not described how the coding was done before the certification system. It may be Iris's text entry mode based on a dictionary. With the introduction of e certification, now they use Iris in the code entry mode. But the primary code assignments to individual causes of death from the drop-down menu may not be the best code choices for assignment.

In Norway, IRIS/ACME has been a fully automated coding system since 2011 and the registration system has remained basically the same, before and after introduction of the online system In the manual system, the full text, as written by the physician, was translated into codes by IRIS. As described in 2), Iris is used in the text entry mode, as a dictionary. When text is selected from the drop-down menu in the electronic reporting system, the chosen diagnoses/conditions appear and are linked to ICD-codes. Thus, the only difference between the manual and the online system is that the manual step of entering written text is bypassed. We hope the revised text in the Introduction and Discussion is clarifying enough. See lines 91-102 and 285-299 in the manuscript.

4. Reduction of ill-defined ICD codes is a positive finding. However, the increase in specific diabetes mellitus codes does not necessarily indicate improved certification and coding practices because the codes are necessarily from the drop-down menu.

As mentioned above, the drop-down menu allows for dictionary searches. We have introduced a discussion on the possibility of position bias, for which we have no indication, in the Discussion. See lines 285-299 in the manuscript.

5. If the study's objective is to investigate the certification practices after e certification system, the authors must conduct a medical record review to validate e death certificate findings with the hospital medical record content. If such a validation proves an increased sensitivity and Positive Predictive Value, it will be a proof of improved cause of death data following e certification.

We fully agree that this would be of great value. However, it would have required a different set-up than in the present study. We already know that death certification is hampered by a certain level of uncertainty, mostly due to lack of training by the physicians or incorrect clinical information*, and instruments have been developed to surpass the most common failures (amongst them Iris/ACME). The purpose of the present study however, was not to quality control the diagnoses listed on the death certificates, but to look for changes in the statistics after introduction of online submission.

*See: Eng HM, L Ellingsen C, Pedersen AG, Alfsen GC. Cause of death certificates in nursing homes: Does quality matter? A retrospective review from two counties in Norway. Scand J Public Health. 2023 Jul 26:14034948231187512. doi: 10.1177/14034948231187512. Epub ahead of print. PMID: 37491994.

6. I would suggest the authors to screen death certificates for the reduction of common errors in medical certification. The most common errors are,

a. Reporting multiple causes per line

b. Absence of time intervals

c. Illegibility

d. Use of non-standard abbreviations

e. Reporting ill-defined causes and modes of dying as underlying cause of death

f. Incorrect causal sequence in part 1 of the death certificates.

A study based on the string of codes after automated analyses, as we have performed, has limitations. Thus, we are not able to analyse for errors regarding the points a- d. 

e) The reporting of ill-defined causes is already part of our study. As we have analysed for deaths by diabetes only, our data do not contain information on deaths registered as by ill-defined causes. However, a recent publication on garbage codes in Norway has interesting information on this*.

f) We have looked at the position of diabetes diagnoses, but only in regard to position I or II, and after automated analyses. A study of incorrect or correct causal sequences would require access to the original death certificates, which we didn’t have.

* see Ellingsen CL, Alfsen GC, Ebbing M, Pedersen AG, Sulo G, Vollset SE, Braut GS. Garbage codes in the Norwegian Cause of Death Registry 1996-2019. BMC Public Health. 2022 Jul 7;22(1):1301. doi: 10.1186/s12889-022-13693-w. PMID: 35794568; PMCID: PMC9261062.

7. The authors can still study the impact on diabetes mortality registration. However, while doing that you may identify the drop-down menu in e certificate as a limiting factor.

We agree and have discussed this limitation. See lines 285-299 in the manuscript.

Reviewer #2: This article addresses an important topic - that of the impact of the introduction of electronic death notification on reporting of diabetes mortality. However there are several serious deficiencies in how the study is presented that need to be addressed if it is to be suitable for publication.

The main issue is that many of the results that are described in the Abstract and the main text - specifically the relative risks from the generalized linear models - are not presented in any tables. 

No Table is referenced in the main text and the results are not presented in any of the tables that are shown. This means that it is not possible to assess the results adequately. Presentation of such results in a table is a pre-requisite for such a manuscript.

We have rewritten parts of the manuscript accordingly. Quality variables have been systematically described in the Materials and Methods and the results collected in a new Table 2, with relevant statistics. See lines 128-141 and 211-230 in the manuscript.

The manuscript does not adequately provide a rationale for the study and the knowledge gap that the study is filling. Why is accurate reporting of diabetes important? For example, what is its ranking as a leading cause of death or disease burden in Norway, what is its impact on the health system? There also needs to be more detailed exploration of the literature on this topic and description of the existing knowledge gap, so that the reader understands the contribution that the manuscript makes.

The mentioned topics are now included in the Introduction. See lines 103-111 in the manuscript.

Is there a variable about region of the country that could be included in the models?

We did not look at regional differences in the present study.

There also needs to be more background about the introduction of the electronic death certificate - how it was rolled-out, how doctors were trained, etc - as well as the process of a doctor completing the electronic death certificate and submitting the certificate.

The Introduction has been expanded accordingly. See lines 74-79 and 80-90 in the manuscript.

What does it mean by "variable for year of death was centered"? This should be clearer.

This is reformulated in the section on statistical method. See lines 150-152 in the manuscript.

The information in Lines 159-163 is not really important and can be deleted or significantly reduced.

We agree. Detailed information on mortality rates regarding cause of death, gender and age has been removed from the text, but can easily be found in the supplementary material. For the same reason, we also have deleted the next sentence, on place of death.

The writing needs to be improved. There are several places with terminology that needs to be more detailed (Line 31 "International classification of diseases 10 codes" should be "10th Revision") or there are errors (Line 36 eCD instead of eDC), I44.9 for cardiac arrest should be I46). Please review carefully.

The manuscript has been reviewed carefully and all errors should be corrected. In regard to I46.9: We are grateful for the thorough reading! It proved out that this was not only a typing error but repeated itself also in our calculations, which is why the results in the new Table 2 now differs slightly from the original manuscript. 

What is "Id" in Line 240?

Id is related to extension of the chain of events. The heading is changed accordingly, to make it more readable. See line 276 in the manuscript.

Line 232: "January 1st2022" should be corrected.

This has been corrected.

Reviewer #3: While commend the efforts of the authors, the following comments are given with the hope that these will be beneficial for them.

Major comments:

1. The introduction section needs to be enriched with more details; more information of the paper-based system used, why countries are shifting to electronic systems, brief account on what is meant by quality of mortality registration and data on diabetes burden (e.g. prevalence) in the Norway.

The Introduction is expanded accordingly, with more information about the death certification systems, mortality registration and disease burden. See lines 74-111 in the manuscript.

2. It would be better to clearly mention and operationalize the variables assessed for quantity and quality in one place in the methodology.

We have systematised all quality variables in the Materials and Methods and collected the results in a new Table 2. See lines 128-141 (and 211-230) in the manuscript.

3. More details are needed in the methods section on how the data analyses were done. As an example how were verification done (e.g. whether two investigators analyzed the data independently etc.)

We have worked with absolute data, and verification procedures as described above, are thus not relevant.

4. It would be better if a table can be inserted to show the relative risk statistics and p values mentioned in abstract and in results.

We agree. See new Table 2. See line217 in the manuscript.

Minor comment

-There is a confusion on the uniformity of the abbreviations used. Examples include:

1. What is meant by "CDs" in the last sentence of the results in abstract

2. eCD and eDC in methods of abstract

Spelling errors have been corrected.

---

## [Decision Letter · Decision Letter 1]

16 Jun 2024

PONE-D-24-05293R1Introduction of Electronic Death Notification in Norway - Impact on Diabetes Mortality RegistrationPLOS ONE

Dear Dr. Eng,

Thank you for submitting your manuscript to PLOS ONE. After careful consideration, we feel that it has merit but does not fully meet PLOS ONE’s publication criteria as it currently stands. Therefore, we invite you to submit a revised version of the manuscript that addresses the points raised during the review process.

We look forward to receiving your revised manuscript.

Kind regards,

Pasyodun Koralage Buddhika Mahesh

Academic Editor

PLOS ONE

Additional Editor Comments:

The authors have taken a commendable effort in addressing the previous comments given by the reviewers. Since one of the previous reviewers was unable to review the revised manuscript, it was sent to new reviewer as well. Authors are advised to address the comments given by this new reviewer along with the additional comments given by the Reviewer 1. In addition authors are advised to:

1. Go through the reference list and ensure that the references are in par with the guidelines of the journal

2. Add a "total" row to the table in S6 file (i.e. in supporting information)

Reviewers' comments:

Reviewer's Responses to Questions

**Comments to the Author**

1. If the authors have adequately addressed your comments raised in a previous round of review and you feel that this manuscript is now acceptable for publication, you may indicate that here to bypass the “Comments to the Author” section, enter your conflict of interest statement in the “Confidential to Editor” section, and submit your "Accept" recommendation.

Reviewer #1: All comments have been addressed

Reviewer #3: All comments have been addressed

Reviewer #4: (No Response)

2. Is the manuscript technically sound, and do the data support the conclusions?

Reviewer #1: Yes

Reviewer #3: Yes

Reviewer #4: (No Response)

3. Has the statistical analysis been performed appropriately and rigorously? 

Reviewer #1: Yes

Reviewer #3: Yes

Reviewer #4: (No Response)

4. Have the authors made all data underlying the findings in their manuscript fully available?

Reviewer #1: Yes

Reviewer #3: Yes

Reviewer #4: (No Response)

5. Is the manuscript presented in an intelligible fashion and written in standard English?

Reviewer #1: Yes

Reviewer #3: Yes

Reviewer #4: (No Response)

6. Review Comments to the Author

Reviewer #1: Thanks for responding to my review questions.

My main concerns were related to the use of a drop-down menu to record diagnoses and assign ICD codes in the eDC and the use of Iris in the code-entry mode for selecting the underlying cause. My comments were based on the following section of the methodology (lines 80 – 88)

‘The processing of DCs by the NCoDR remains unchanged. DCs completed from the drop-down menu are automatically assigned with ICD-codes and in most causes do not require further handling. Causes of death in pDCs og eDCs with free text are manually registered. The resulting string of codes reflects the order of diseases or conditions given in the DC, regardless of type of submission and/or use of free text or drop-down menu. The string of codes from each DC is subject to automatic processing by an automated coding system (Iris) based on the Automatic Classification of Medical Entry software (ACME), which selects UCOD according to rules and guidelines provided by WHO (ICD-10) (10, 11, 12). Complex cases are reviewed by specially trained staff from the NCoDR before selection of UCOD in Iris.’

As I explained before, the use of drop-down menus and the selection of ICD codes from the drop-down can lead to systematic errors in certification and coding.

However, the resubmission explains the certification and coding process (Lines 91-100), and the clarity has been improved.

‘ After submission, the processing of DCs by the NCoDR remains unchanged. eDCs completed from the drop-down menu are automatically assigned with ICD-10 codes. Allocations of ICD-codes in death certificates with free text are achieved after manual registration of diagnoses, by means of handling by the automated coding system (Iris) based on the Automatic Classification of Medical Entry software (ACME) (13, 14). The resulting “string of codes” reflects the order of diseases or conditions given in the DC, regardless of type of submission and/or use of free text or drop-down menu. The string of codes from each DC is subject to renewed automated processing by Iris/ACME, which selects UCOD according to rules and guidelines provided by WHO (ICD-10) ( 15). Automated processing may lead to changes or specifications of the original ICD-10 code on the DC, dependent on the setting of diagnoses on the DC.’

In your response to reviewer question 6, you mentioned, ‘A study of incorrect or correct causal sequences would require access to the original death certificates, which we didn’t have.’ I would appreciate it if you could explain the difference between eDC and the original death certificate. How could the original death certificate differ from eDC? For what purposes do you use the original death certificate?

Reviewer #3: The author has sufficiently and clearly addressed the comments provided in the previous review. The introduction is now more coherent, effectively emphasizing the significance of transitioning to the electronic death certificate system in Norway. The quality of the manuscript has been greatly improved and I extend my congratulations to the authors.

Reviewer #4: •This study conducted in Norway examines the impact of introducing the electronic death certification system in 2018 and making e-notifications mandatory in 2022. The authors compared the diagnosis of diabetes as the underlying cause of death (UCOD) from 2017 to 2022, finding an increase in reporting diabetes as UCOD after these changes. The study's rationale is clear and valid, highlighting the importance of improving diabetes management in Norway. The authors employed a suitable protocol and feasible methodology to effectively achieve the study's aims. I commend the authors for this important study following changes in the death certification system. However, I have the following comments:

•Since 5% of submissions in 2022 were paper-based, it is important to mention how many of these were diabetes-related. If the percentages of paper-based diabetes-related deaths in 2017 and 2022 are similar, it would suggest that the sample is representative across both years and submission methods. The authors should also clearly clarify whether the 5% paper-based diabetes-related deaths in 2022 were excluded from the analysis((127).

•Minor correction in line 59: “freetext” should be corrected to "free text" if it is not one word.

•In line 100, define what T1DM stands for (Type 1 Diabetes Mellitus).

•There are several errors in Table 2 (line 217). Although it mentions -N (%), only numbers are provided without percentages. The position of UCOD in part I, RR: 1.03 CI: 1.00-1.05, should be checked for significance, as a CI including 1 is not significant. Additionally, the mean number of diagnoses per DC and the corresponding RR is not mentioned. Results of Table 2 are not adequately interpreted.

•There was no targeted training of physicians in connection with the introduction of the new submission system (line 76). It can be included as a recommendation to provide such training to further improve the system.

7. PLOS authors have the option to publish the peer review history of their article (what does this mean?). If published, this will include your full peer review and any attached files.

Reviewer #1: No

Reviewer #3: **Yes: **I.O.K.K.Nanayakkara

Reviewer #4: **Yes: **W.D.J.K Amarasena

---

## [Author Response · Author response to Decision Letter 1]

27 Jul 2024

To the editors.

Regarding PONE-D-24-05293R1

We thank you for thorough reviews with valuable comments, undoubtedly contributing to raise the quality and ease the readability of our manuscript.

Changes in the present (and hopefully final) revision are listed below: 

1. The reference list has been reviewed in detail and should now be in par with the guidelines of the journal.

2. A "total" row to the table in S6 file (i.e. in supporting information) is added.

Comments from reviewers:

Reviewer #1: Thanks for responding to my review questions.

My main concerns were related to the use of a drop-down menu to record diagnoses and assign ICD codes in the eDC and the use of Iris in the code-entry mode for selecting the underlying cause. My comments were based on the following section of the methodology (lines 80 – 88)

‘The processing of DCs by the NCoDR remains unchanged. DCs completed from the drop-down menu are automatically assigned with ICD-codes and in most causes do not require further handling. Causes of death in pDCs og eDCs with free text are manually registered. The resulting string of codes reflects the order of diseases or conditions given in the DC, regardless of type of submission and/or use of free text or drop-down menu. The string of codes from each DC is subject to automatic processing by an automated coding system (Iris) based on the Automatic Classification of Medical Entry software (ACME), which selects UCOD according to rules and guidelines provided by WHO (ICD-10) (10, 11, 12). Complex cases are reviewed by specially trained staff from the NCoDR before selection of UCOD in Iris.’

As I explained before, the use of drop-down menus and the selection of ICD codes from the drop-down can lead to systematic errors in certification and coding.

However, the resubmission explains the certification and coding process (Lines 91-100), and the clarity has been improved.

‘ After submission, the processing of DCs by the NCoDR remains unchanged. eDCs completed from the drop-down menu are automatically assigned with ICD-10 codes. Allocations of ICD-codes in death certificates with free text are achieved after manual registration of diagnoses, by means of handling by the automated coding system (Iris) based on the Automatic Classification of Medical Entry software (ACME) (13, 14). The resulting “string of codes” reflects the order of diseases or conditions given in the DC, regardless of type of submission and/or use of free text or drop-down menu. The string of codes from each DC is subject to renewed automated processing by Iris/ACME, which selects UCOD according to rules and guidelines provided by WHO (ICD-10) ( 15). Automated processing may lead to changes or specifications of the original ICD-10 code on the DC, dependent on the setting of diagnoses on the DC.’

We are pleased to see that this section now is satisfactory.

In your response to reviewer question 6, you mentioned, ‘A study of incorrect or correct causal sequences would require access to the original death certificates, which we didn’t have.’ I would appreciate it if you could explain the difference between eDC and the original death certificate. How could the original death certificate differ from eDC? For what purposes do you use the original death certificate?

As explained in the introduction (lines 88-89), the physicians have the possibility of free text also in the electronic version of DCs. Certificates with free text require manual handling. Thus, the “string of codes” after handling by the NCoDR may not be fully identical to the submitted, original version, even in eDCs. Unfortunately, data on how often free text is used in electronic submitted DCs is presently not recorded. 

We fully agree that studies on the occurrence of incorrect causal sequence in part 1 of the death certificates would be an excellent quality measure. A complicating factor is that the IT-system used in NCoDR does not allow for the detection of which WHO-rules are used when selecting UCOD. We believe that analyses as suggested in question 6-f also will require a larger material than diabetes deaths only. We are presently planning to look into this, using data from all deaths. We hope to present these data by next year.

Reviewer #3: The author has sufficiently and clearly addressed the comments provided in the previous review. The introduction is now more coherent, effectively emphasizing the significance of transitioning to the electronic death certificate system in Norway. The quality of the manuscript has been greatly improved and I extend my congratulations to the authors.

We are grateful for constructive comments to our manuscript.

Reviewer #4: •This study conducted in Norway examines the impact of introducing the electronic death certification system in 2018 and making e-notifications mandatory in 2022. The authors compared the diagnosis of diabetes as the underlying cause of death (UCOD) from 2017 to 2022, finding an increase in reporting diabetes as UCOD after these changes. The study's rationale is clear and valid, highlighting the importance of improving diabetes management in Norway. The authors employed a suitable protocol and feasible methodology to effectively achieve the study's aims. I commend the authors for this important study following changes in the death certification system. However, I have the following comments:

•Since 5% of submissions in 2022 were paper-based, it is important to mention how many of these were diabetes-related. If the percentages of paper-based diabetes-related deaths in 2017 and 2022 are similar, it would suggest that the sample is representative across both years and submission methods. The authors should also clearly clarify whether the 5% paper-based diabetes-related deaths in 2022 were excluded from the analysis((127). 

The numbers of death certificates on paper for all deaths are given in table 1 (2022: n=1995 / 4,7% of all deaths). For diabetes, relations of diabetes deaths and type of death certificate submission are illustrated in Figure 1, and the data given in supporting information S2 (pDC 2022: n=29 / 3,7% of diabetes deaths). As discussed in lines 200-205, this difference was significant.

Due to the small number of pDCs in 2022, we did not exclude diabetes deaths reported on paper in 2022 (2022: 29/ 1995=1.45 %) We have added a sentence in Statistical methods explaining this (see line 154).

•Minor correction in line 59: “freetext” should be corrected to "free text" if it is not one word. 

Corrected.

•In line 100, define what T1DM stands for (Type 1 Diabetes Mellitus). 

As this is an introductory example, abbreviations have been removed.

There are several errors in Table 2 (line 217). Although it mentions -N (%), only numbers are provided without percentages. 

We apologize for having forgotten to add percentages. These are now added.

The position of UCOD in part I, RR: 1.03 CI: 1.00-1.05, should be checked for significance, as a CI including 1 is not significant. 

We are grateful for this important correction and have changed the manuscript accordingly (lines 48-49 (abstract), table 2, line 221 and 225 (results), and line 253 (Key results)).

Additionally, the mean number of diagnoses per DC and the corresponding RR is not mentioned. Results of Table 2 are not adequately interpreted.

Under statistical methods we explain the different methods used for the different quality variables. For variable 4, where data were normally distributed, we used a two-sample t-test, and linear regressions to estimate additive changes and CI. To clarify this, we have revised the language under statistical methods to “a two-independent sample t-test” instead of “a two-sample t-test”. See line 158.

•There was no targeted training of physicians in connection with the introduction of the new submission system (line 76). It can be included as a recommendation to provide such training to further improve the system.

The online cause of death submission system introduced in Norway was thought to be intuitive, requiring no further training. However, we fully agree that training systems for improving cause of death registrations by physicians should be introduced. We have suggested to the Norwegian health authorities to make the existing online course mandatory. We have added a recommendation on this in the discussion (line 275-278).

Attached: 

Manuscript with and without changes (“Revised Manuscript with Track Changes” and “Manuscript”)

Figure 1 (no changes)

Supporting information (S1 File-S5 File (no changes) and S6 File added with a "total" row)

We hope that the answers to the reviewers’ questions and that the revision of the script as well are satisfactory.

Best regards, 

Hanna Maria Eng

---

## [Decision Letter · Decision Letter 2]

13 Aug 2024

PONE-D-24-05293R2Introduction of Electronic Death Notification in Norway - Impact on Diabetes Mortality RegistrationPLOS ONE

Dear Dr. Eng,

Thank you for submitting your manuscript to PLOS ONE. After careful consideration, we feel that it has merit but does not fully meet PLOS ONE’s publication criteria as it currently stands. Therefore, we invite you to submit a revised version of the manuscript that addresses the points raised during the review process.

We look forward to receiving your revised manuscript.

Kind regards,

Pasyodun Koralage Buddhika Mahesh

Academic Editor

PLOS ONE

Journal Requirements:

Reviewers' comments:

Reviewer's Responses to Questions

**Comments to the Author**

1. If the authors have adequately addressed your comments raised in a previous round of review and you feel that this manuscript is now acceptable for publication, you may indicate that here to bypass the “Comments to the Author” section, enter your conflict of interest statement in the “Confidential to Editor” section, and submit your "Accept" recommendation.

Reviewer #1: All comments have been addressed

Reviewer #4: All comments have been addressed

2. Is the manuscript technically sound, and do the data support the conclusions?

Reviewer #1: Yes

Reviewer #4: Partly

3. Has the statistical analysis been performed appropriately and rigorously? 

Reviewer #1: Yes

Reviewer #4: Yes

4. Have the authors made all data underlying the findings in their manuscript fully available?

Reviewer #1: Yes

Reviewer #4: Yes

5. Is the manuscript presented in an intelligible fashion and written in standard English?

Reviewer #1: Yes

Reviewer #4: Yes

6. Review Comments to the Author

**Reviewer #1: **Thanks for submitting your responses to the reviewer questions.

While accepting you submission, I would like to draw your attention to some of your responses. In your response, you have said that all death certificates completed using free-text require manual handling. That is not correct. I'm sure you use a Iris dictionary for the purpose of batch processing in the text-entry mode. Iris is able to code death certificates containing free text in batch processing.

You have also mentioned that your IT system doesn't indicate which rules were used during mortality coding. However, Iris MUSE (Multi Causal and Unicausal Selection Engine) provides the users a full explanation of coding rules applied and the entire coding process.

**Reviewer #4:** Thank you for addressing the previous comments. Please find the revised comments below:

1. Line 143:

o Comment: Please ensure that abbreviations are not used at the beginning of sentences. The revised text should fully spell out terms to enhance clarity and readability.

2. Table 2, Line 218:

o Comment: The confidence interval (CI) for Underlying Cause of Death (UCOD) listed as 1-1.05 is not statistically significant. Please review and correct this information to accurately reflect statistical significance.

o

7. PLOS authors have the option to publish the peer review history of their article (what does this mean?). If published, this will include your full peer review and any attached files.

Reviewer #1: No

Reviewer #4: **Yes: **W.D.J.K Amarasena

---

## [Author Response · Author response to Decision Letter 2]

2 Sep 2024

To the editors.

Regarding PONE-D-24-05293R1

We thank you for thorough reviews with valuable comments, undoubtedly contributing to raise the quality and ease the readability of our manuscript.

Comments from reviewers:

Reviewer #1: Thanks for submitting your responses to the reviewer questions.

While accepting you submission, I would like to draw your attention to some of your responses. In your response, you have said that all death certificates completed using free-text require manual handling. That is not correct. I'm sure you use a Iris dictionary for the purpose of batch processing in the text-entry mode. Iris is able to code death certificates containing free text in batch processing. We are grateful for these comments. To clarify this, we have chosen to rephrase this part in the manuscript (line 91-102 in Manuscript with Track Changes).

You have also mentioned that your IT system doesn't indicate which rules were used during mortality coding. However, Iris MUSE (Multi Causal and Unicausal Selection Engine) provides the users a full explanation of coding rules applied and the entire coding process. We are sorry that our dataset did not allow for this type of statistics.

Reviewer #4: Thank you for addressing the previous comments. Please find the revised comments below:

1. Line 143:

o Comment: Please ensure that abbreviations are not used at the beginning of sentences. The revised text should fully spell out terms to enhance clarity and readability.

Corrected (line 147 in Manuscript with Track Changes).

2. Table 2, Line 218:

o Comment: The confidence interval (CI) for Underlying Cause of Death (UCOD) listed as 1-1.05 is not statistically significant. Please review and correct this information to accurately reflect statistical significance.

For the record, statistical analyses were run again. 

In this regard, we discovered a typing error from the previous run. We apologize for this and are grateful for this correction.

The result is unchanged in terms of statistical significance (p>0.05).

The correct results are RR=1.10, CI: 0.96-1.26, p=0.152. 

We have corrected the values in table 2. 

We hope that the answers to the reviewers’ questions and the revision of the script are satisfactory.

Attached: 

Manuscript with and without changes (“Revised Manuscript with Track Changes” and “Manuscript”)

Figure 1 and supporting information (S1 File-S6 File) (no changes)

Best regards, 

Hanna Maria Eng

---

## [Decision Letter · Decision Letter 3]

13 Sep 2024

Introduction of Electronic Death Notification in Norway - Impact on Diabetes Mortality Registration

PONE-D-24-05293R3

Dear Dr. Eng,

We’re pleased to inform you that your manuscript has been judged scientifically suitable for publication and will be formally accepted for publication once it meets all outstanding technical requirements.

Kind regards,

Pasyodun Koralage Buddhika Mahesh

Academic Editor

PLOS ONE

Additional Editor Comments (optional):

Reviewers' comments:

Reviewer's Responses to Questions

**Comments to the Author**

1. If the authors have adequately addressed your comments raised in a previous round of review and you feel that this manuscript is now acceptable for publication, you may indicate that here to bypass the “Comments to the Author” section, enter your conflict of interest statement in the “Confidential to Editor” section, and submit your "Accept" recommendation.

Reviewer #1: All comments have been addressed

Reviewer #4: All comments have been addressed

2. Is the manuscript technically sound, and do the data support the conclusions?

Reviewer #1: Yes

Reviewer #4: Yes

3. Has the statistical analysis been performed appropriately and rigorously? 

Reviewer #1: Yes

Reviewer #4: Yes

4. Have the authors made all data underlying the findings in their manuscript fully available?

Reviewer #1: Yes

Reviewer #4: Yes

5. Is the manuscript presented in an intelligible fashion and written in standard English?

Reviewer #1: Yes

Reviewer #4: Yes

6. Review Comments to the Author

Reviewer #1: The document is a manuscript draft discussing the impact of electronic death notification on diabetes mortality registration in Norway. All reviewer comments have been successfully addressed by the authors.

Reviewer #4: I would like to extend my sincere gratitude for your efforts in revising the manuscript. The corrections have been done carefully and all suggested changes have been addressed appropriately. 

7. PLOS authors have the option to publish the peer review history of their article (what does this mean?). If published, this will include your full peer review and any attached files.

Reviewer #1: **Yes: **U S H GAMAGE

Reviewer #4: **Yes: **W.D.J.K Amarasena

---

## [Editor Report · Acceptance letter]

25 Sep 2024

PONE-D-24-05293R3 

PLOS ONE

Dear Dr. Eng, 

I'm pleased to inform you that your manuscript has been deemed suitable for publication in PLOS ONE. Congratulations! Your manuscript is now being handed over to our production team.

Kind regards, 

on behalf of

Dr. Pasyodun Koralage Buddhika Mahesh 

Academic Editor

PLOS ONE